# Segmenting Moving Objects via an Object-Centric Layered Representation

**Junyu Xie**[1]       **Weidi Xie**[1,2]       **Andrew Zisserman**[1]

[1]Visual Geometry Group, Department of Engineering Science, University of Oxford, UK
[2]Coop. Medianet Innovation Center, Shanghai Jiao Tong University, China
{jyx,weidi,az}@robots.ox.ac.uk

## Abstract

The objective of this paper is a model that is able to discover, track and segment multiple moving objects in a video. We make four contributions: First, we introduce an object-centric segmentation model with a depth-ordered layer representation. This is implemented using a variant of the transformer architecture that ingests optical flow, where each query vector specifies an object and its layer for the entire video. The model can effectively discover multiple moving objects and handle mutual occlusions; Second, we introduce a scalable pipeline for generating multi-object synthetic training data via layer compositions, that is used to train the proposed model, significantly reducing the requirements for labour-intensive annotations, and supporting Sim2Real generalisation; Third, we conduct thorough ablation studies, showing that the model is able to learn object permanence and temporal shape consistency, and is able to predict amodal segmentation masks; Fourth, we evaluate our model, trained only on synthetic data, on standard video segmentation benchmarks, DAVIS, MoCA, SegTrack, FBMS-59, and achieve state-of-the-art performance among existing methods that do not rely on any manual annotations. With test-time adaptation, we observe further performance boosts.

## 1   Introduction

Humans have the ability to discover and segment moving objects in videos. This ability is present even in young infants, together with notions of object permanence through occlusions, and temporal shape constancy [4, 5, 50, 54]. Achieving this ability by machine has long been a goal of the field [7–9, 14, 27, 33, 45, 58, 59, 65, 67, 68], and modern methods are now able to reliably discover and segment single moving objects using both appearance (RGB) and motion (optical flow) streams of the video. However, success is limited in the case of multiple moving objects and occlusion.

Our goal in this paper is to develop and study a model that can discover multiple moving objects and also handle their mutual occlusions. To this end, we propose a *layered* object-centric model that predicts *amodal* segmentation masks [73] for each object throughout the video. Note that, given amodal segmentations, the depth ordering of the object layers can be determined from occlusions in the observed (modal) frames when the objects overlap. See Figure 1. Our innovation is to build this mechanism into the representation: when objects overlap in a frame, the model has to maintain the amodal shapes through temporal consistency, and can thus infer depth ordering.

To achieve this goal we use only optical flow as our primary information stream. There are benefits and drawbacks to this choice. The benefits are two fold: first, the information that we are after, the segmentation masks, are directly available from discontinuities in the flow field; second, a key advantage of using solely optical flow is that the domain adaptation problem is largely avoided [15, 32] – and this means that the model can be trained on synthetic sequences and applied directly to real sequences without a significant Sim2Real disparity. However, the drawbacks are also two fold: first,

36th Conference on Neural Information Processing Systems (NeurIPS 2022).

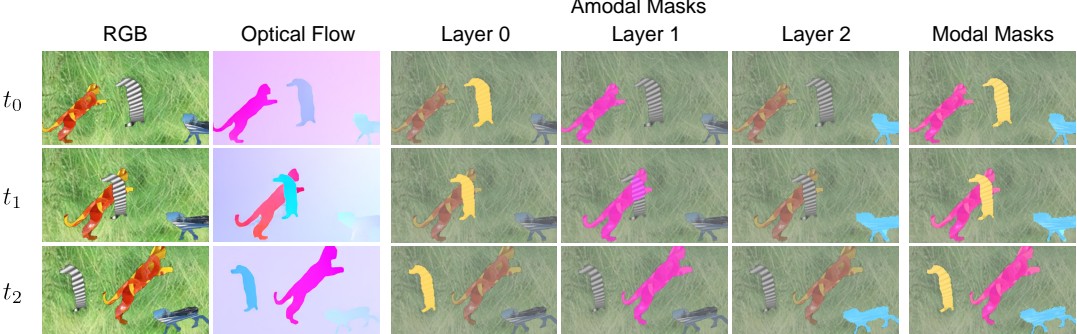

Figure 1: **Multiple-object Discovery and Segmentation.** We develop a layered model for segmenting and tracking multiple objects by their motion, even under mutual occlusion. The model takes only optical flow as its input, and predicts an **amodal** segmentation (whole physical structure) for each object, and associating an object to the same layer throughout the video. The resultant **modal** masks (visible parts) can be constructed by overlaying layers based on the inferred depth ordering (in this case, layer 0-1-2 from front to back). From left to right: RGB sequence, estimated (imperfect) optical flows, amodal mask inferences from our model and resultant modal masks.

unlike the appearance stream, objects can be invisible in flow when they are not moving, thereby requiring the model to build a concept of object permanence; and second, but more subtly, flow is not like appearance in that we cannot expect constancy even when the object is moving (for example, due to camera shake that affects the flow but not the appearance), and thus the model cannot strongly use flow coherence as a cue but must instead learn to model the segmentation mask shape.

We make four contributions: (i) we introduce an **O**bject-**C**entric **L**ayered **R**epresentation (**OCLR**) model for discovering and segmenting multiple moving objects and inferring their mutual occlusions from optical flow alone. The architecture of the model is based on a transformer, with DETR like learnable queries, where each query determines the amodal segmentation mask for an object and its layer order throughout the video. (ii) we introduce a layer-based synthetic pipeline to generate multi-object training sequences that cover the variability of the target real sequences in terms of: multiple deforming objects, overlapping objects in some frames, moving cameras, and non-static backgrounds. (iii) We demonstrate that the model trained on flow from the synthetic videos is able to learn object permanence and temporal shape constancy – even if an object is partially occluded in many frames, the model is still able to predict an amodal mask. (iv) Finally, we evaluate the model (trained only on synthetic sequences) on multiple real video segmentation benchmarks, DAVIS, MoCA, SegTrack, and FBMS-59; and demonstrate strong Sim2Real generalisation, achieving superior segmentation performance compared to other human-label-free methods. The addition of a test-time adaptation to include appearance from a self-supervised DINO model boosts the performance further, even outperforming some supervised approaches that are finetuned on real data sequences.

## 2 The Object-Centric Layered Representation (OCLR) Model

This section describes our object-centric layered representation model that ingests optical flow for the frames of a video, and outputs the segmentation masks and tracks of the moving objects it contains. The model also determines the number of moving objects and their depth order. Specifically, the model predicts amodal (*i.e.* complete) segmentation masks and associates each object with a layer. The layers are composed to form the modal segmentations in the observed optical flow. The model is trained on synthetic sequences to learn temporal coherence and maintain the object shape even under occlusion, and as a consequence, develops a notion of object permanence.

### 2.1 Layered representation

For a given video with $T$ frames, we can represent its motion by optical flows, $\mathcal{V}_{\text{flow}} = \{F_1, F_2, \ldots, F_T\}$, we aim to segment the moving objects in videos by adopting a layered model, where each object is segmented and consistently associated to one layer throughout the video:

$$\{(\hat{\mathcal{A}}_1, \hat{r}_1), \ldots, (\hat{\mathcal{A}}_T, \hat{r}_T)\} = \Phi(\mathcal{V}_{\text{flow}}) \tag{1}$$

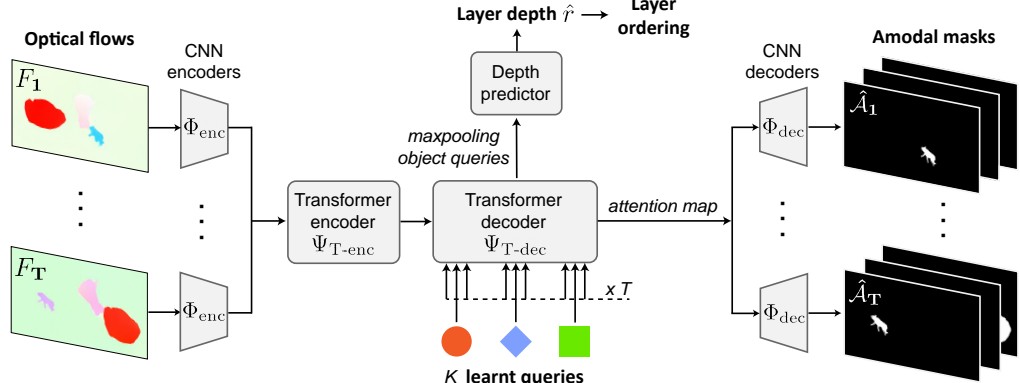

Figure 2: **OCLR Model architecture.** The model is a U-Net architecture with a Transformer bottleneck. It takes optical flow as input and extracts spatial features by CNN encoders. A transformer encoder jointly processes spatio-temporal features across all frames, followed by a transformer decoder that determines the layer representations. Each learnable query vector in the transformer decoder is associated to one object and used to infer its layer depth. Additionally, the cross-attention maps from the last transformer decoder are extracted and upsampled by CNN decoders to infer amodal segmentation for the moving objects. Note, the skip connections from the CNN encoders to decoders are not shown.

where $\hat{\mathcal{A}}_i \in \mathbb{R}^{H \times W \times K}$ refers to the binary **amodal** segmentation of the $i$-th frame, together with a predicted layer depth $\hat{r}_i \in \mathbb{R}^K$. $H, W$ refer to the input height and width, and $K$ denotes the number of object layers. Note that, the layer depth value here does not have a true meaning, but only indicates the layer ordering. By definition, at all time steps, the same layer would remain empty or always bind to the **same** object throughout the video, *i.e.* objects can be effectively tracked by the considered layered representation.

**Modal mask generation.** Given the predicted amodal segmentation and layer depth, we describe the composition procedure to generate the **modal** segmentation for moving objects. In specific, we permute the index of layers (**amodal** segmentations) based on their depth values, such that, $\hat{\mathcal{A}}^{k-1}$ is always in front of $\hat{\mathcal{A}}^k$. Note that, pixels in amodal masks defined here are binary-valued, *i.e.* either 0 or 1. For the $i$-th frame, its modal segmentation can thus be computed via an iterative, front-to-back blending procedure :

$$\hat{\mathcal{M}}_i^k = (1 - \alpha_i^k) \odot \hat{\mathcal{A}}_i^k \tag{2}$$

where $\alpha_i^k = \text{clip}(\alpha_i^{k-1} + \hat{\mathcal{M}}_i^{k-1}, 0, 1)$, denoting an accumulated opacity layer with $\alpha_i^0 = 0$.

## 2.2 Architecture

In order to accommodate the layered representation, we introduce an architecture that takes optical flows as input, and outputs a set of layers and their depths, see Figure 2. We adopt a U-Net architecture with a Transformer-based bottleneck, the queries in transformer decoder play the role as layers in our proposed representation, each is either empty or encoding the amodal shape of one moving object along the video.

**CNN backbone.** Given a sequence of optical flow inputs, a U-Net encoder is used to compute frame-wise features, $\mathcal{V} = \{v_1, \dots, v_T\} = \{\Phi_{\text{enc}}(F_1), \dots, \Phi_{\text{enc}}(F_T)\}$, where $v_i \in \mathbb{R}^{h \times w \times c}$ refers to the feature maps, with height $h$, width $w$ and number of feature channels $c$. Following [67], we use Instance Normalisation (IN) [60] after each convolution, which encourages the separation of foreground motions from background.

**Transformer encoder.** We flatten the outputs from the ConvNets in both temporal and spatial dimensions, and inject the positional information by adding a set of learnable spatio-temporal embeddings. As a consequence, the video features are converted into a sequence of vectors, and passed into Transformer Encoder to model the temporal relationship across multiple frames.

**Transformer decoder.** We pass the vector sequence to transformer decoder, where $K$ learnable embeddings are used as object queries for individual layers. We broadcast the queries along the

temporal dimension, and supply positional information with learnable temporal embeddings, $\mathcal{Q}_{obj} \in \mathbb{R}^{TK \times c}$. The queries that only differ in temporal embeddings will be associated to the same moving object along the video.

As outputs from transformer decoder, we extract the last cross-attention maps and recover their spatial dimensions, $\rho \in \mathbb{R}^{TK \times h \times w \times \text{nheads}}$, which will be progressively upsampled with skip connections from the ConvNets encoders, resulting in binary amodal segmentation for the object queries in each frame, $\{\hat{\mathcal{A}}_1, \ldots, \hat{\mathcal{A}}_T\} = \Phi_{\text{dec}}(\rho)$.

**Layer ordering.** Apart from amodal segmentations, the model will also infer the depth of the layers. We assume a global layer ordering that remains fixed across frames. As shown in Figure 2, output object queries from transformer decoder are maxpooled along the temporal dimension and passed through two feed-forward layers, to get the output $\hat{r} \in \mathbb{R}^{1 \times K}$, indicating the depth for each of the $K$ layers. Intuitively, this is equivalent to reason the layer depth based on one selected key occlusion frame from the whole sequence. By defining the lowest depth value as corresponding to the top layer, we then obtain a depth order prediction between layers. Note that, for layers that are empty or contain non-interacting objects, the ordering is not strictly defined, thus any prediction should not be penalised.

### 2.3  Training objectives

In this section, we describe the training procedure for the proposed architecture, with groundtruth supervision on both amodal segmentation and layer ordering.

**Amodal segmentation loss.** In addition to the conventional pixelwise binary classification ($\mathcal{L}_{\text{bce}}$) on each of the $K$ layers, we also adopt a pixelwise classification loss defined only on a strip region around object boundaries, termed as the boundary loss $\mathcal{L}_{\text{bound}}$. This additional loss further emphasises the boundary regions, forcing the model to maintain shape information. During training, since the predicted amodal segmentations are permutation invariant, we use Hungarian matching to match their corresponding ground truth [12], and then compute the amodal loss :

$$\mathcal{L}_{\text{amodal}} = \frac{1}{KT} \sum_{k=1}^{K} \sum_{i=1}^{T} \left( \lambda_{\text{bce}} \cdot \mathcal{L}_{\text{bce}}(\hat{\mathcal{A}}_i^k, \mathcal{A}_i^k) + \lambda_{\text{bound}} \cdot \mathcal{L}_{\text{bound}}(\hat{\mathcal{A}}_i^k, \mathcal{A}_i^k) \right) \tag{3}$$

where $\lambda_{\text{bce}}$ and $\lambda_{\text{bound}}$ are loss weights.

**Layer ordering loss.** For layers with mutual occlusions, we train the order prediction module with ranking losses on all $K!$ pairs of layers, formally,

$$\mathcal{L}_{\text{order}} = \frac{1}{K!} \sum_{i \neq j} -\log\left( \sigma\left( \frac{\hat{r}_i - \hat{r}_j}{\tau} \right) \right) \tag{4}$$

where $\sigma$ is the sigmoid function with a temperature factor $\tau$, and the ground-truth ordering indicates a relative depth relationship $\hat{r}_i > \hat{r}_j$ between layer $i$ and $j$.

**Total loss.** The overall loss for training is a combination of amodal $\mathcal{L}_{\text{amodal}}$ and ordering loss $\mathcal{L}_{\text{order}}$:

$$\mathcal{L}_{\text{total}} = \mathcal{L}_{\text{amodal}} + \lambda_{\text{order}} \cdot \mathcal{L}_{\text{order}} \tag{5}$$

where $\lambda_{\text{order}}$ is the weight factor for the layer ordering loss.

**Discussion.** As discussed in early sections, our model is trained to infer amodal (*i.e.* complete) object segmentations and layer depth, even though the object itself may sometimes be invisible in the flow field due to being temporarily static, or partially occluded by other objects. As a result, the model is forced to always maintain the object shape internally, *i.e.* a notion of object permanence, as well as infer mutual occlusions from the visible flows. In the next section, we will introduce a pipeline for simulating video sequences, where groundtruth amodal segmentations and layer orderings can be generated at scale to train our model.

## 3  Synthetic dataset generation

We introduce a scalable pipeline to synthesize videos to train the proposed layered object-centric model. The pipeline builds on the method of Lamdouar *et al.* [32], but extends it to simulating videos with multiple objects and complex inter-object occlusions through a layer composition.

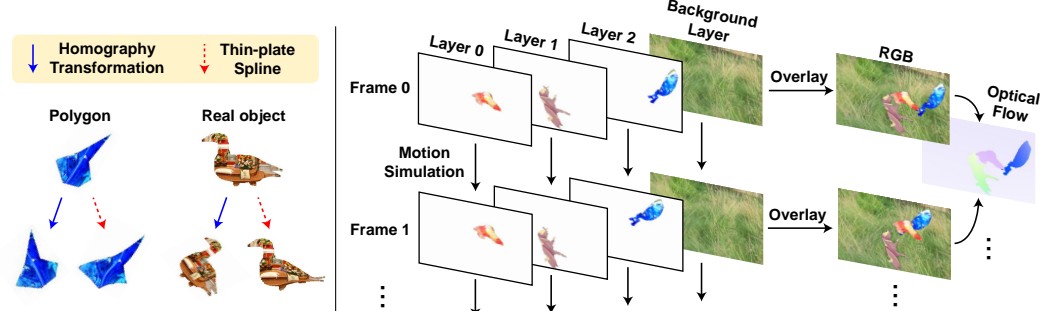

Figure 3: **Overview of the synthetic data pipeline.** Left: Motion simulations of polygon and real object sprites based on homography transformations (including spatial translations) and thin-plate spline. Right: Layer composition of frames with independently moving objects.

As shown in Figure 3, we first simulate independent motions for foreground objects and backgrounds within each layer. To construct a frame, we overlay the layers via an iterative binary blending procedure, and an occlusion occurs when objects in different layers reside in the same spatial position, similar to the idea discussed in Sect. 2.1. The proposed pipeline yields rich information at frame level, including RGB appearances, groundtruth optical flows, and modal and amodal object segmentations. Additionally, we can also compute optical flows for the RGB sequences with an off-the-shell flow estimator, for example, RAFT [57]. In the following sections, we describe some key procedures for video simulation, and leave more technical details to the Supplementary Material.

**Foreground objects.** We generate opaque 2D sprites with random shapes and RGB textures as foreground objects. In detail, we adopt two major shapes, namely polygons, and real object sprites. The polygon shapes can be either convex or non-convex, with different numbers of sides. A minimum distance between vertices is also defined to better support vertex-based deformations. On the other hand, the real object masks are directly sourced from the silhouettes in the YouTubeVOS dataset [66]. These generated objects are then applied with textures sampled from the PASS dataset [3], which is a large-scale image dataset with human identifiable information removed.

**Non-rigid object transformations.** To simulate non-rigid motions, we apply both homography transformations and thin plate splines to the objects. Thin plate splines define a non-linear coordinate transformation with a set of control points. For polygon-shaped objects, we consider their vertices as control points, and spatially perturb a subset of the vertices by distances $d_i \in (0, D_i/2)$, where $D_i = \min(|\mathbf{P}_i - \mathbf{P}_j|, \forall j)$, defines the minimum distance between the perturbed point $\mathbf{P}_i$ and any other points $\mathbf{P}_j$. For real object silhouettes, a uniform grid is distributed over the object, and the grid points are defined as control points, with $D_i$ being the distance between adjacent grid points.

**Backgrounds and artificial occluders.** To simulate the camera motions, we randomly sample images from the PASS dataset, and apply random homography transformations and color jitterings. Additionally, we replace backgrounds with real videos cropped from copyright-free sources to simulate the complex background motions and depth variations. For cases with objects being occluded, we introduce artificial occluders superimposed as the top layer. These occluders are designed to have the same motion as the background.

**Summary.** The proposed simulation pipeline enables an arbitrary number of videos to be generated, together with multiple types of groundtruth annotation; for example, amodal segmentation and layer ordering. We prepare 4k synthetic sequences (around 120k frames) for training, the video sequences contain 1, 2, or 3 objects in equal proportions. In the following, we train all models on this synthetic video dataset unless otherwise specified.

## 4 Experiments

### 4.1 Datasets

To evaluate our multi-layer model, we benchmark on several popular datasets for video object segmentation tasks. A brief overview of the datasets is given below, with full details in the Supplementary Material.

For single object video segmentation, we evaluate the model on DAVIS2016 [49], SegTrackv2 [34], FBMS-59 [46] and MoCA [33]. Note that, despite the fact that multiple objects may be annotated, the community often treats SegTrackv2 and FBMS-59 as a benchmark for single object segmentation [25, 68] by grouping objects in the foreground.

To benchmark motion-based segmentation for multiple objects, we introduce a synthetic validation dataset (Syn-val) and a curated dataset (DAVIS2017-motion). The former is generated with the same parameters as our synthetic training set (Sect. 3), containing over 300 multi-object sequences (around 10k frames) with $1, 2, 3$ objects at equal proportions, and controllable occlusions for evaluating modal and amodal segmentations in the ablation studies. Moreover, as objects in common motion cannot be distinguished purely from motion cues, we re-annotate the original DAVIS2017 [51] dataset by grouping jointly moving objects to form a new DAVIS2017-motion dataset. We will release our modified annotations for future research.

## 4.2 Evaluation metrics

Following the common practice [49], we adopt region similarity ($\mathcal{J}$), *aka.* intersection-over-union, in single object segmentation tasks. While for multiple object segmentation, we additionally consider the contour similarity ($\mathcal{F}$) [44]. As for MoCA dataset, since only bounding box annotations are provided, we follow the same metric used in [37, 67], and report a mean object detection success rate averaged over different IoU thresholds $\{0.5, 0.6, 0.7, 0.8, 0.9\}$.

## 4.3 Implementation details

**Training details.** As for pre-processing, we use RAFT to estimate optical flows at $\pm 1$ frame gaps, and resize the obtained flows to $128 \times 224$. During training, we split the video sequences into 30 frames per sample, each input frame is first encoded by a U-Net encoder into a feature map with $1/16$ of its original spatial resolution, and passed to the transformer bottleneck. We use $K = 3$ learnable object queries, associating to 3 independent foreground layers. The model is trained by the Adam optimizer [28] with a learning rate linearly warmed up to $5 \times 10^{-5}$ during 40k iterations, and decreased by half every 80k iterations.

**RGB-based test-time adaptation.** To alleviate the drawbacks from using purely flow information (*e.g.* stationary objects over a long period of time), we investigate the benefits of test-time adaptation with appearance features. For this we use the self-supervised DINO-pretrained vision transformer [13] (ViT-S/8, patch sizes $8 \times 8$). The model is adapted by using the per-frame object masks obtained from the flow-based model predictions as noisy annotations. These annotations are used to finetune the last two layers of the vision transformer by supervised contrastive learning.

The finetuned model for each sequence is then applied for mask propagation in the same manner as [24]. Instead of propagating from the first frame of the sequence, we formulate a heuristic selection process that picks a single key frame based on temporal coherence between OCLR segmentation predictions. Object masks in this selected frame are then bi-directionally propagated across the whole sequence. Apart from the starting frame information, flow-predicted masks in other frames are also selectively introduced during the mask propagation process as a form of dynamic refinement. Finally, we adopt CRF as a post-processing step to refine the resultant mask predictions. More technical details can be found in the Supplementary Material.

## 4.4 Ablation study

Here, we present a series of ablation studies on training details and pipeline for data simulation however, due to the space limitation, we can only summarise some key findings, we refer the reader to Supplementary Materials for all details.

As shown in Table 1, we can make the following observations: *First*, Instance Normalisation in CNN encoder is indispensable for training the network, as indicated by Ours-A vs. Ours-C; *Second*, training on only modal mask degrades model performance on all datasets, especially on Syn-Val with multiple objects and occlusions, suggesting that explicit amodal supervision are critical for learning object permanence, as indicated by Ours-B vs. Ours-D; *Third*, a longer temporal input (Ours-C vs. Ours-D) tends to result in slightly higher overall performance; *Fourth*, applying boundary loss (Ours-C vs. Ours-E) leads to a noticeable performance boost. This validates our assumption that

Table 1: **Settings for training parameters. IN**: Instance Normalisation; **Amodal**: Training on amodal mask (vs. modal mask); $\lambda_{\text{bound}}$: weight on boundary loss; $T$: number of input frames. Syn-Val ($\mathcal{M}|\mathcal{A}$) corresponds to modal and amodal results on synthetic dataset.

| | Training Settings | | | | $\mathcal{J}$ (Mean) $\uparrow$ | | |
|---|---|---|---|---|---|---|---|
| Model | IN | Amodal | $\lambda_{\text{bound}}$ | $T$ | Syn-Val ($\mathcal{M}\mid\mathcal{A}$) | DAVIS2016 | DAVIS2017-motion |
| Ours-A | ✗ | ✓ | 0.2 | 30 | 83.5 \| 83.0 | 67.6 | 48.7 |
| Ours-B | ✓ | ✗ | 0.2 | 30 | 81.1 \| 76.9 | 69.2 | 50.5 |
| Ours-C | ✓ | ✓ | 0.2 | 30 | **85.6 \| 84.7** | **72.1** | **54.5** |
| Ours-D | ✓ | ✓ | 0.2 | 15 | 82.8 \| 83.0 | 71.3 | 53.5 |
| Ours-E | ✓ | ✓ | 0 | 30 | 80.9 \| 81.6 | 71.5 | 54.1 |

focusing on object boundaries can help the model to learn the information regarding object shapes and layer orders from optical flows.

## 4.5 Single object video segmentation

In this section, we compare our model with state-of-the-art methods on various single object segmentation benchmarks. Note that, we mainly compare with the self-supervised approaches, as both lines share the same spirit in the sense that training *does not* require any manual annotation, nor fine-tuning on the target dataset. As shown in Table 2, our flow-only model demonstrates superior performance over all other human-label-free approaches. Figure 4 provides qualitative illustrations of the model, in comparison with other state-of-the-art methods. It can be seen that our predictions are temporally consistent and not affected by noticeable background distractors in flow signals. Furthermore, our model is capable of handling complex scenarios including heavy object deformations and occlusions. The other methods are not able to maintain the object shape consistently, and consequently their performance is weaker.

Table 2: Quantitative comparison on single object video segmentation benchmarks. "HA" stands for human annotations, and "SR" refers to the detection success rate on MoCA. In column Sup. (supervision), "None", "Syn.", "Real" represent self-supervision, synthetic data supervision, and real data supervision, respectively. ***Bold*** represents the state-of-the-art performance (excluding our test-time adaptation results, which are labelled as *blue* instead).

| | Training Settings | | | | $\mathcal{J}$ (Mean) $\uparrow$ | | | SR (Mean) $\uparrow$ |
|---|---|---|---|---|---|---|---|---|
| Model | HA | Sup. | RGB | Flow | DAVIS2016 | SegTrackv2 | FBMS-59 | MoCA |
| NLC [17] | ✗ | None | ✓ | ✓ | 55.1 | 67.2 | 51.5 | − |
| CIS (w. post-process.) [68] | ✗ | None | ✓ | ✓ | 71.5 | 62.0 | 63.5 | 0.363 |
| Motion Grouping [67] | ✗ | None | ✗ | ✓ | 68.3 | 58.6 | 53.1 | 0.484 |
| SIMO [32] | ✗ | Syn. | ✗ | ✓ | 67.8 | 62.0 | − | 0.566 |
| **OCLR (flow-only)** | ✗ | Syn. | ✗ | ✓ | **72.1** | **67.6** | **65.4** | **0.599** |
| **OCLR (test. adap.)** | ✗ | Syn. | ✓ | ✓ | *80.9* | *72.3* | *69.8* | 0.559 |
| FSEG [25] | ✓ | Real | ✓ | ✓ | 70.7 | **61.4** | 68.4 | − |
| COSNet [42] | ✓ | Real | ✓ | ✗ | 80.5 | 49.7 | 75.6 | 0.417 |
| MATNet [72] | ✓ | Real | ✓ | ✓ | 82.4 | 50.4 | **76.1** | **0.544** |
| D$^2$Conv3D [53] | ✓ | Real | ✓ | ✗ | **85.5** | − | − | − |

The RGB-based test-time adaptation gives a further performance boost on most of the benchmarks, and is even competitive to some supervised approaches that have been finetuned on the target video data. Note that the test-time adaptation is actually detrimental to performance on MoCA. This is not unexpected though, as MoCA has many camouflage sequences where the objects are not visually distinguishable in appearance from their background environment, and motion provides crucial cues for discovering them.

## 4.6 Multiple object video segmentation

To the best of our knowledge, no existing unsupervised approaches have reported performance segmenting multiple objects purely based on optical flow. Apart from re-running the original self-supervised Motion Grouping [67] method (with three foreground queries) as a baseline on

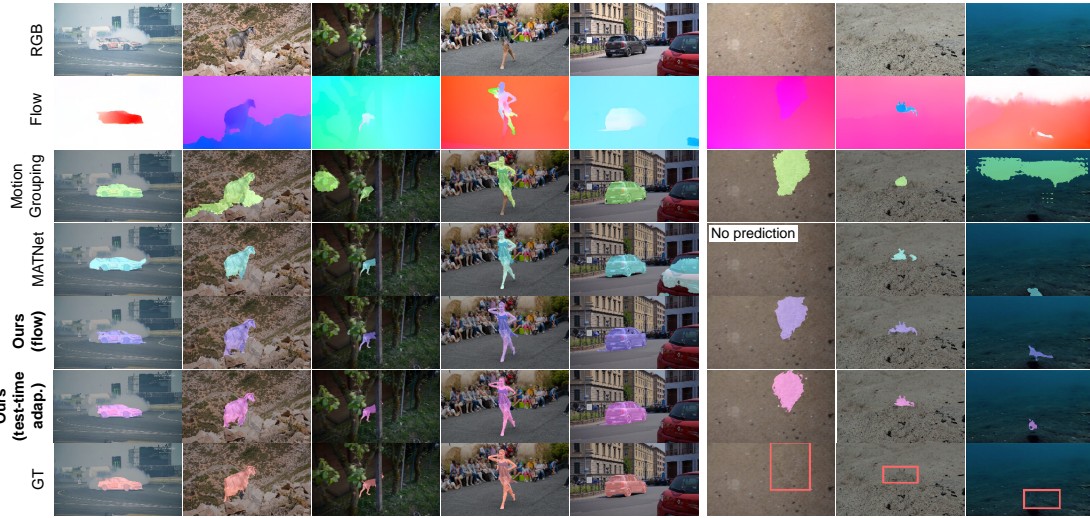

Figure 4: Qualitative results of single object video segmentation on DAVIS2016 and MoCA.

DAVIS2017-motion, we also train a variant of the Motion Grouping model directly supervised by our synthetic dataset, namely Motion Grouping (sup.). As an additional baseline, we train a Mask R-CNN [22] model with only optical flow inputs on our synthetic data, and refer to this as Mask R-CNN (flow-only). We also compare with existing semi-supervised approaches, where the first-frame ground-truth segmentation is provided, and the model only needs to propagate the masks through the video, thus an easier task than ours which requires to simultaneously discover the objects and track them.

As shown in Table 3, our proposed flow-only model outperforms both the Motion Grouping and Mask R-CNN baselines. This is because our OCLR model exploits a layered representation to maintain the object shape through the video, which enables to segment the objects that are under occlusion or having unnoticeable motion. In contrast, Motion Grouping only predicts per-frame segmentations, *i.e.* not forced to preserve temporal relations between objects. This conjecture can also be confirmed by the qualitative results in Figure 5, for example, despite the person and one dog (4th column) are not visible in the flow, our model still correctly recovers them. Moreover, after RGB-based test-time adaptations, the performance can be further boosted both quantitatively and qualitatively, for example, the boundary of the pig mask in 1st column, the person in the 5th column.

Table 3: Quantitative comparison of multi-object video segmentation on DAVIS2017-motion. Note that, the compared methods here are trained without using any human annotations during training, in particular, Motion Grouping (sup.), Mask R-CNN (flow-only) and OCLR models are supervised by only synthetic data, and other approaches are trained with self-supervision. ***Bold*** represents the state-of-the-art performance (excluding our test-time adaptation results, which are labelled as *blue* instead).

| | Training settings | | | DAVIS2017-motion performance | | |
|---|---|---|---|---|---|---|
| Model | 1st-frame-GT | RGB | Flow | $\mathcal{J}\&\mathcal{F}\uparrow$ | $\mathcal{J}$ (Mean) $\uparrow$ | $\mathcal{F}$ (Mean) $\uparrow$ |
| Motion Grouping [67] | ✗ | ✗ | ✓ | 35.8 | 38.4 | 33.2 |
| Motion Grouping (sup.) | ✗ | ✗ | ✓ | 39.5 | 44.9 | 34.2 |
| Mask R-CNN (flow-only) | ✗ | ✗ | ✓ | 50.3 | 50.4 | 50.2 |
| **OCLR (flow-only)** | ✗ | ✗ | ✓ | **55.1** | **54.5** | **55.7** |
| **OCLR (test. adap.)** | ✗ | ✓ | ✓ | 64.4 | 65.2 | 63.6 |
| CorrFlow [31] | ✓ | ✓ | ✗ | 54.0 | 54.2 | 53.7 |
| UVC [36] | ✓ | ✓ | ✗ | 65.5 | 66.2 | 64.7 |
| MAST [30] | ✓ | ✓ | ✗ | 70.9 | 71.0 | 70.8 |
| CRW [24] | ✓ | ✓ | ✗ | 73.4 | 72.9 | 74.1 |
| DINO [13] | ✓ | ✓ | ✗ | **78.7** | **77.7** | **79.6** |

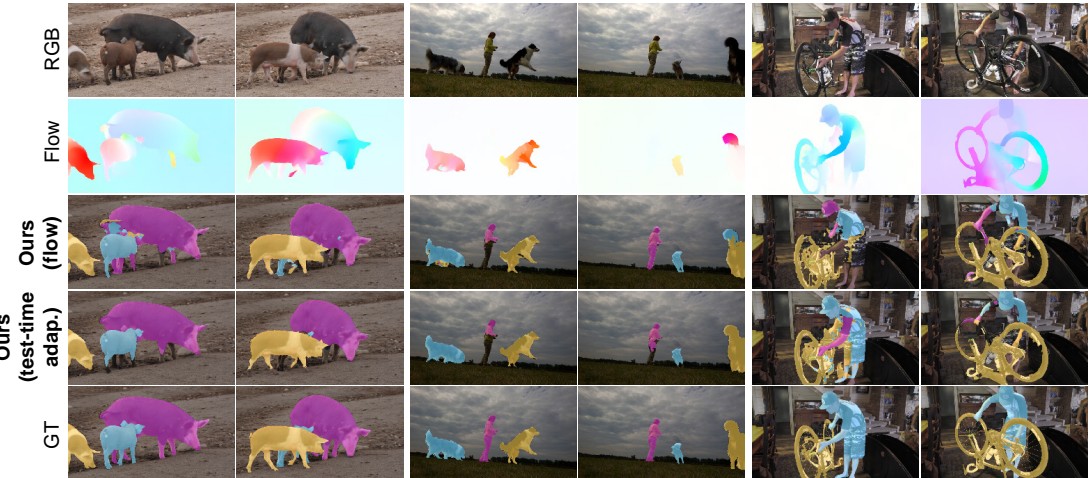

Figure 5: Qualitative results of multi-object video segmentation on DAVIS2017-motion.

# 5 Related Work

**Video object segmentation** has been a longstanding task in computer vision, two protocols have attracted increasing interest in the recent literature [6, 9, 11, 14, 18, 19, 25–27, 30, 31, 43, 45, 47, 48, 51, 59, 61, 62, 66, 69], namely, semi-supervised video object segmentation (**semi-supervised VOS**), and unsupervised video object segmentation (**unsupervised VOS**). The former aims to re-localize one or multiple targets that are specified in the first frame of a video with pixel-wise masks, and the latter considers automatically segmenting the object of interest (usually the most salient one) from the background in a video sequence. Despite being called **unsupervised VOS**, in practice, the popular methods address such problems by supervised training on large-scale external datasets, this is in contrast to our proposed approach that does not rely on human annotations whatsoever.

**Motion segmentation** focuses on discovering the *moving* objects in videos. In the literature, [9, 27, 45, 65] proposed to cluster the pixels with similar motion patterns; [14, 58, 59] train deep networks to map the motions to segmentation masks. In [68], adversarial training was adopted to leverage the independent motions between the moving object and its background; In [7, 8, 33], the authors propose to highlight the independently moving object by compensating for the background motion, either by registering consecutive frames, or explicitly estimating camera motion. In [67], a Transformer-like architecture is used to reconstruct the input flows, and the segmentation masks can be generated as a side product. In contrast to the existing approaches that primarily focus on single moving object segmentation, we focus on segmenting **multiple** moving objects, even under mutual occlusions.

**Layered representation** was originally proposed by Wang and Adelson [63], to represent a video as a composition of layers with simpler motions. Recently, the layer decomposition ideas have been adopted for novel view synthesis [55, 71], separating reflections and other semi-transparent effects [1, 2, 20, 40, 41], or foreground/background estimation [20]. Unlike these approaches that primarily focus on graphics applications, we propose a layered representation to handle the occlusion problems in multi-object discovery purely from their motions.

**Object-centric representation** decomposes the scenes into "objects", normally, visual frame reconstruction has been widely used as training objective, for example, IODINE [21] uses iterative variational inference to infer a set of latent variables recurrently, with each representing one object in an image. Similarly, MONet [10] and GENESIS [16] also adopt multiple encoding-decoding steps, [29, 39] propose a slot attention mechanism, which enables the iterative binding procedure. In this paper, we also adopt a Transformer variant, but focus on discovering objects in videos by motions.

**Amodal segmentation** refers to segmenting the complete shape of objects, including the invisible parts due to possible occlusions. Some existing approaches address this problem by applying human-estimated [52, 56, 64] or synthetic [23] supervision, while other works generate training datasets with synthetic occluders [35, 38, 70]. Although not specifically targeting the amodal segmentation task, our work leverages the idea of amodal perception and grants the trained model a notion of object permanence via synthetic amodal mask supervision.

# 6 Discussion

We have achieved our design goal, in that the model is able to handle multiple objects, and their occlusions, and to predict depth ordering of their layers. Also, the model demonstrates superior performance, both quantitatively and qualitatively, over prior methods that rely on zero human annotations. Nevertheless, there are of course some limitations and room for further improvements: *First*, the current method may fail in very challenging real-world scenarios such as heavy object deformations, complex mutual interactions, etc. Probably a more sophisticated data simulation pipeline with highly articulated objects or even 3D sprites would help to further reduce this Sim2Real domain gap. *Second*, our test-time adaptation has demonstrated a remarkable performance improvement benefiting from the combination between flow inferences and RGB correlations. This overcomes, to an extent, the problem of the flow-only model 'forgetting' the shape of objects when they are stationary for multiple frames. A further direction could naturally be to incorporate RGB information into our flow-based network in pre-training. *Third*, the model currently uses a global depth ordering for the sequence, so it cannot handle situations such as order altering and mutual occlusions. Potential future studies could work on a more sophisticated layered model by focusing on pair-wise relationships between objects.

Despite these limitations, the approach has convincingly demonstrated both the value of inferring amodal segmentation masks, in order to handle occlusions, and the possibility of training such models entirely on synthetic sequences.

**Acknowledgements**

This research is funded by EPSRC Programme Grant VisualAI EP/T028572/1, a Royal Society Research Professorship RP\R1\191132, and a Clarendon Scholarship. We thank Tengda Han for proof-reading.

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
