# OpenReview forum: "Segmenting Moving Objects via an Object-Centric Layered Representation"
_NeurIPS.cc/2022/Conference — NeurIPS 2022 Accept_

### Official Review · Reviewer_2xHR · 2022-07-01

**Rating:** 6
**Confidence:** 4
**Soundness:** 3 good
**Presentation:** 4 excellent
**Contribution:** 3 good

**Summary:**

This paper proposes a U-net for generating a motion segmentation of a given motion field. In particular the method assumes T snapshots of the field and generates T amodal segmentation masks. The novelty of the paper lies in combining the U-net with a Transformer, i.e. the Transformer receives the latent embeddings of the U-Net encoder and generates output embeddings from which the segmentations masks and an ordering of these masks is inferred. The authors follow the idea of layered motion and assume that each motion segment has unique depth by that the ordering is interpreted as a depth ordering. The model is limited to a fixed size of layers (queries), in this paper three layers. The method is then trained on new synthetic flow data without any manual labelling and applied to a variety of existing datasets and also to new data.

**Questions:**

How would your method behave in the case of multiple persons walking frontoparallel in the same direction?
Is your method influences by the choice of the optical flow method?
You assume three layers / three motion segments. How did you come to this assumption of three?
What is your idea how to handle this dilemma between appearance and flow for camouflaged cases in your method (line 238)?

**Limitations:**

Yes.

**Strengths And Weaknesses:**

The paper is very well written, very well structured. The methodological approach sounds appropriate and the presentation of the results is well done. I am not working on this specific problem but I believe the authors that the proposed neural network model is novel and not published before. It shows how this generative problem can be elegantly solved by combining a U-Net for the task of segmentation together with a Transformer for the task of inferring spatiotemporal correlations to guide the segmentation. The evaluation seems properly done with partially very good results, although some statements go to far, e.g. in line 227 where it is argued “…large margin …”, although improvements are not that large e.g.for FBMS-59.
I believe that it has been shown that motion segmentation is basically learnable from synthetic data only. If so, one of the co-reviewers more expert than I will definitely refer to this.
What I miss in the paper is a clear distinction between motion segmentation and occlusion reasoning (2.1D sketch). As the model can only infer motion boundaries, it might miss some of the occlusion boundaries especially for surfaces with interior occlusion boundaries. It is also not clear to the reader how the model behaves for corrupted flow inputs or for more than three motions, e.g. also for a moving camera. Finally, the reader might also be interested to understand why longer sequences give better results or why object boundaries can help improving the layer ordering (line 223).
Monet is also a new method to infer occlusion regions and motion boundaries (H. Kim et al., BMVC’21). I suggest to clarify this in the paper. I also suggest to add X. Zhan et al.’s, CVPR’20 paper to the related work that shows a nice way to compute amodal maps.

---

> ### Author Response · Authors · 2022-08-02
> **Response to Reviewer 2xHR (Part 1)**
>
> Thanks for your review. Please see our comments below specifically addressing the weaknesses and questions. Owing to the length limit, we have split the response into two parts.
>
> **Q1: ''The evaluation seems properly done with partially very good results, although some statements go to far, e.g. in line 227 where it is argued “…large margin …”, although improvements are not that large e.g.for FBMS-59.''**
>
> A1: Thanks for the comment, we have modified some of our descriptions regarding performance comparisons in the revised version.
>
> **Q2:  ''What I miss in the paper is a clear distinction between motion segmentation and occlusion reasoning (2.1D sketch). As the model can only infer motion boundaries, it might miss some of the occlusion boundaries especially for surfaces with interior occlusion boundaries.''**
>
> A2: The transformer-based architecture enables our model to maintain the temporally consistent predictions of amodal object shapes. In other words, shape information in other frames is utilized to recover the complete shape of objects being occluded.
>
>
> **Q3: ''It is also not clear to the reader how the model behaves for corrupted flow inputs or for more than three motions, e.g. also for a moving camera.''**
>
> A3: Our model is trained on synthetic data with RAFT-predicted flows rather than perfect GT flows, thus, the model is trained to extract correct object shapes from occasional erroneous flow signals, via exploiting temporal consistency between frames. This can be verified by the results in Table E3 in the supplementary material, where the model trained on non-perfect RAFT flows (Ours-B) outperforms the GT-flow-trained counterpart (Ours-A) on all real dataset benchmarks.  Fig. 4 also provides some examples of corrupted flows. In the 4th column, the flow signal of the dancer’s leg is very weak, while, in the 7th column, the left part of the camouflaged crab disappears in the flow fields, our model successfully recovers the missing parts from inaccurate flow inputs.
>
> As introduced in Section 3, we also simulate camera motions in our synthetic dataset by either applying homography motions to the background image at every frame or adopting a moving real-video background. Qualitative results demonstrate that our model can correctly extract the moving object under large camera motion. For example, in the 3rd column of Fig. 4, also in the 1st and 3rd column of Fig. G8, our model successfully captures objects such as the dog, racing cars and the running girl, unaffected by rapid camera motions.
>
>
> **Q4: ''Finally, the reader might also be interested to understand why longer sequences give better results or why object boundaries can help improve the layer ordering (line 223). Monet is also a new method to infer occlusion regions and motion boundaries (H. Kim et al., BMVC’21). I suggest to clarify this in the paper.''**
>
> A4: Benefits from applying a longer sequence and emphasizing object boundaries are separately discussed below.
>
> With a greater time span of the input flows, predictions of object shapes and positions can be improved by referring to more frames through the attention mechanism in the Transformer-based bottleneck. Moreover, by comparing through longer temporal information, short-term noises in optical flows owing to, for example, glaring and water splashing, could also be averaged out, as they are not temporally consistent over a long term.
>
> Applying an extra loss on object boundaries (both for visual parts and invisible parts) forces the network to focus on the amodal shape of objects and better learn object permanences based on temporal information. With an improved prediction of complete object shapes, it would be simpler for the network to learn the layer orders by comparing the predicted amodal masks with apparent object shapes revealed by flow signals.
>
> As suggested, MONet [1] predicts both occlusion and motion boundaries based on appearance and bi-directional flow inputs, which is inspirational for our future study. Owing to limited space, we will add this paper to the related work in the camera-ready version where one additional page is allowed.
>
> ### Reference
> [1] Kim et al. "Joint Detection of Motion Boundaries and Occlusions".  In *BMVC*, 2021

---

> ### Author Response · Authors · 2022-08-02
> **Response to Reviewer 2xHR (Part 2)**
>
>
> Please find the second part of our comments below.
>
> **Q5: ''I also suggest to add X. Zhan et al.’s, CVPR’20 paper to the related work that shows a nice way to compute amodal maps.''**
>
> A5: Thanks for the suggestion. PCNet [1] proposed in X. Zhan et al.’s, CVPR’20 paper takes in single images and completes both amodal shapes and textures for occluded objects with an elegant graph-based layer order representation. We will add this work to the related work in the camera-ready version, as it is inspirational for potential further investigations on a more sophisticated layer representation and amodal completions based on video appearance.
>
>
> **Q6: ''How would your method behave in the case of multiple persons walking frontoparallel in the same direction?''**
>
> A6: For the case when multiple people are very close to each other and share similar motion trends, our model may struggle in separating these people into different layers. This problem falls into the category of “common motion” as introduced in Section D.2 and cannot be simply resolved based on only optical flow inputs. We regard this common motion problem as one of the limitations of our current network and consider addressing it as potential further work.
>
>
> **Q7: ''Is your method influenced by the choice of the optical flow method?''**
>
> A7: Yes, as our network directly takes in optical flow estimations as inputs, from which object shape, position and motion information is extracted. Below shows an extra ablation study investigating the performance as a result of various optical flow methods, which also verifies our choice of RAFT method for optical flow estimations.
>
> | Flow   |  DAVIS16 IoU | DAVIS17-motion IoU|
> | :--------: | :--------: | :--------: |
> | ARFlow [2] | 54.0 | 39.5 |
> | MaskFlownet [3] | 66.0  | 49.0  |
> | RAFT [4]| **72.1** | **54.5** |
>
> We noticed that recent work such as FlowFormer [5] shows more advanced flow estimation performance and will consider applying these methods for performance updates and further work.
>
>
> **Q8: ''You assume three layers / three motion segments. How did you come to this assumption of three?''**
>
> A8: We would like to first mention that our OCLR model is layer-agnostic, i.e., extendable to a different number of layers by varying the number of learnable queries in the transformer decoder. Reasons to adopt three layers can be discussed from two perspectives: (i) Existing video segmentation benchmarks commonly include less than or equal to three objects for each video. To better evaluate our model performance, we chose to align our settings to these benchmark settings. (ii) Admittedly, a few benchmarks involve more objects per sequence (e.g., 10 objects). We reckon that motion-based segmentation methods at the current stage are not applicable to too many moving objects, as the resultant optical flow patterns would be very chaotic. On the other hand, we believe that segmenting three layers from purely optical flow inputs is already a sufficient challenge that has not been approached before.
>
>
> **Q9: ''What is your idea how to handle this dilemma between appearance and flow for camouflaged cases in your method (line 238)?''**
>
> A9: Our method applies the OCLR model to process only flow information, followed by a separate test-time adaption procedure that depends solely on RGB appearance information. As explained in line 238, this later stage at test time does not help with the particular case of camouflaged detection. To tackle this problem, a possible future direction would be to introduce RGB information into the model during training, such that the model can learn to internally extract and combine the most useful information from both optical flows and appearances (e.g., sharp motion boundaries in flows, distinguishable textural information in appearances.)
>
>
>
>
> ### Reference
> [1] Zhan et al. "Self-supervised scene de-occlusion".  In *CVPR*, 2020.
>
> [2] Liu et al. "Learning by Analogy: Reliable Supervision from Transformations for Unsupervised Optical Flow Estimation".  In *CVPR*, 2020.
>
> [3] Zhao et al. "MaskFlownet: Asymmetric Feature Matching with Learnable Occlusion Mask".  In *CVPR*, 2020.
>
> [4] Teed et al. "RAFT: Recurrent All Pairs Field Transforms for Optical Flow".  In *ECCV*, 2020.
>
> [5] Huang et al. "FlowFormer: A Transformer Architecture for Optical Flow". In *ECCV*, 2022.

---

### Official Review · Reviewer_mEda · 2022-07-11

**Rating:** 6
**Confidence:** 4
**Soundness:** 4 excellent
**Presentation:** 3 good
**Contribution:** 3 good

**Summary:**

This paper considers the problem of segmeting multiple moving objects in a scene by taking optical flow as inputs. The solution is composed of two key ideas: First, the inputs are optical flow, so they can employ simulated videos for training, and the learned model is able to generalize to real videos. Second, a depth-oredered layered representation is used to handle mutual occlusion. Experimental results show improvements upon methods with non-layered representation on single/multiple object segmentation datasets.

**Questions:**

There might be some hard cases for layered representation. In some cases, objects have mutual occlusion with each other, say a part of object A occludes object B, meanwhile a part of object B occludes object A. Can the proposed method handle such cases? If not, is there any solutions to such cases under the layered representation framework?

**Limitations:**

The authors have discussed the limitations of their work.

**Strengths And Weaknesses:**

Originality:

Good. The key innovations are three fold: (1) object-centric representation; (2) layered representation; (3) amodel representation. Although all three techniques have already been explored in previous literature, there is no such an attempt before to combine them for video object segmentation. Furthermore, most relevant works experiment on simulated datasets, while in this work all experiments are conducted on real world videos, which is much more difficult and more convincing.

Quality:

Good. The proposed method is easy to understand and shows good performance on all dataset considered, surpassing the most relevant method, Motion Grouping, by a significant margin.

Clarity:

Good but can be improved. In general the paper reads smoothly and is easy to follow. But, perhaps due to limited space, some important details are defered to the supplementary material, which from my point of view, would be better to be placed in the main text. For example, test-time adaptation seems bring large improvements and is important to the final performance, it would be clearer to put relevant details in the main text.

Significance:

The sutdied topic is of great importance. Object-centric, layered repersentation shows promise to become the next generation of vision paradigm.

---

> ### Author Response · Authors · 2022-08-02
> **Response to Reviewer mEda**
>
> Thanks for your review and the comment on significance. Please see our comments below specifically addressing the weaknesses and questions.
>
> **Q1: ''Test-time adaptation seems to bring large improvements and is important to the final performance, it would be clearer to put relevant details in the main text.''**
>
> A1: This was due to limited space, with details on the test-time adaptation process provided in the supplementary material. If the paper is accepted, then in the camera-ready version, we will aim to bring a more complete explanation of test-time adaptation into the main text where an additional page is allowed.
>
>
>
> **Q2: ''There might be some hard cases for layered representation. In some cases, objects have mutual occlusion with each other, say a part of object A occludes object B, meanwhile a part of object B occludes object A. Can the proposed method handle such cases?  If not, are there any solutions to such cases under the layered representation framework?''**
>
> A2: We agree that complex object interactions such as mutual occlusions are challenging problems which cannot be modelled with depth-ordered layers. However, the current situations in real data are already very challenging, and we have succeeded in segmenting real-world data with this simple depth-ordered representation  (and using only flow inputs).
> We will add this model limitation to the list and discuss it in the paper. To address the problem, future work could focus on predicting occlusion relationships between every pair of layers/objects (rather than predicting an ordering). This can be implemented, for example, by a dynamic graph representation with occlusions indicated by directed edges as proposed in [1]. Under this representation, mutual occlusions between A and B can be simply represented by two directed edges pointing in-between nodes A and B.
>
> ### Reference
> [1] Zhan et al. "Self-supervised scene de-occlusion".  In *CVPR*, 2020.

---

> > ### Comment · Reviewer_mEda · 2022-08-10
> > **Post rebuttal comments**
> >
> > Thanks for the rebuttal!
> > My concern regarding the test-time adaptation is well-addressed. Also, I agree with the authros that mutual occlusion is a challenging problem, and it is not necessary to be solved in this work as the current contributions are sufficient. Therefore, I maintain my positive rating.

---

### Official Review · Reviewer_FcJQ · 2022-07-11

**Rating:** 5
**Confidence:** 4
**Soundness:** 2 fair
**Presentation:** 3 good
**Contribution:** 3 good

**Summary:**

The authors propose a new model for amodal segmentation of videos. Given optical flow,
a Transformer-based neural network predicts an amodal (i.e., unoccluded) segmentation
mask and depth for each object in the input frame. The model is trained using ground
truth supervision on a synthetic dataset created by the authors. When evaluated on
common video instance segmentation benchmarks, the model shows promising results without
requiring further training. The proposed model clearly outperforms previous unsupervised
segmentation methods and even outperform some supervised methods trained on the
respective benchmarks.

**Questions:**

What is your notion of an "unsupervised" model? It is stated that the proposed model
"achieve[s] state-of-the-art unsupervised segmentation performance". The model however
is trained with ground truth supervision, so in my view the model cannot be termed
unsupervised and the comparison to truly unsupervised methods is misleading. Instead, it
should be made clearer that the proposed method does "unsupervised transfer".

I cannot completely follow the computation of the model segmentation:
- The first accumulated opacity layer $\alpha^0$ should probably be all zeros. As it is
  stated now, it follows that $\alpha^1=2\cdot \hat{M}^0$.
- The modal segmentation masks are added to obtain the opacity layers. This however
  means that in the case of occlusions it might be $\alpha^k > 0$, leading to the
  amodal mask being multiplied with a negative weight. Did you take any means to adress
  this problem?

**Limitations:**

One limitation of the model arises due to using optical flow as primary input, which
makes it impossible to segment non-moving objects. This limitation is discussed openly
by the authors and subjected to future work, what I believe is justified.

**Strengths And Weaknesses:**

Video object segmentation is an active area of research. Predicting layered amodal segmentation masks that can be combined into a modal
segmentation is an uncommon but very natural approach. With this approach, the model is tasked to not
only learn about visible object parts but complete objects. Occlusions arise naturally
when masks are combined. Compared to the more common approach of only reasoning about
visible object fragments this may result in a more useful object representation.

The main weakness of the paper in my view is the comparison with other models. The
proposed model is trained supervisedly on synthetic data and then transferred without
supervision. None of the other models is trained in this way, but either completely
unsupervised or trained supervisedly on the respective benchmark. The performance
improvements on the benchmark are therefore not necessarily due to the proposed model
architecture, but might as well arise due to the different data seen during training.

A fair comparison would be to also train the other methods on the synthetic dataset and
evaluate on the benchmarks. The key question then would be whether the transfer
performance of the proposed model is better than that of previous models, which would be
especially interesting for other models using optical flow.

---

> ### Author Response · Authors · 2022-08-02
> **Response to Reviewer FcJQ**
>
> Thanks for your review. Please see our comments below specifically addressing the weaknesses and questions.
>
>
>
> **Q1: ''What is your notion of an "unsupervised" model? It is stated that the proposed model "achieve[s] state-of-the-art unsupervised segmentation performance". The model however is trained with ground truth supervision, so in my view the model cannot be termed unsupervised and the comparison to truly unsupervised methods is misleading.''**
>
> A1: Please refer to the response to Q1 in the overall response, where we explain the reasons for adopting the term “unsupervised” and comparing our results with other unsupervised methods.
>
> **Q2: ''A fair comparison would be to also train the other methods on the synthetic dataset and evaluate the benchmarks.''**
>
> A2: To the best of our knowledge, there are only a limited number of models tackling multi-object motion segmentation problems with optical flows as the only inputs. To provide an additional ablation, we modify the self-supervised Motion Grouping method [1] by applying direct synthetic supervision, with all other network settings remaining unchanged. The table below suggests that our OCLR model outperforms Motion Grouping supervised by the same synthetic dataset (Syn-train).
>
> |  Model  | DAVIS17-motion IoU|
> | :--------: | :--------: |
> | Motion Grouping (Syn.) [1] | 44.9  |
> | OCLR (flow-only) | **54.5** |
>
>
> On the other hand, it should be highlighted that the multi-object synthetic generation pipeline is also one of the important contributions made by our work. Both synthetic data generation and the OCLR model based on flow are indispensable factors in our Sim2Real approach.
>
> **Q3: ''The first accumulated opacity layer $\alpha^0$ should probably be all zeros. As it is stated now, it follows that $\alpha^1 = 2 \hat{M}^0$.''**
>
> A3: Yes. Thank you for pointing this out. There is an omission in the definition of the initial state, which would be correct only if a clipping between 0 and 1 is applied on every accumulated mask $\alpha^k$. (see the explanation of clipping in the next response, A4). The correct definition is $\alpha^0 = 0$ and $\hat{M}^0 = \hat{A}^0$.
>
> **Q4: ''The modal segmentation masks are added to obtain the opacity layers. This however means that in the case of occlusions it might be $\alpha^k>0$, leading to the amodal mask being multiplied with a negative weight. Did you take any means to address this problem?''**
>
> A4: Sorry for the confusion. We should have made it clearer that amodal masks are first hard-thresholded (by 0.5 to give either 0 or 1) to give opaque object layers before modal mask constructions. Since the modal mask construction process is not involved in training and back-propagations, there is no need to stick to soft-valued amodal masks. During the layer accumulation stage, opacity mask values $\alpha^k$ are always **clipped between 0 and 1**, so negative modal mask values would never occur. We will add these missing details to the corresponding sections.
>
> ### Reference
> [1] Yang et al. "Self-supervised Video Object Segmentation by Motion Grouping". In *ICCV*, 2021

---

> > ### Comment · Reviewer_FcJQ · 2022-08-06
> > **Re: Response to Reviewer FcJQ**
> >
> > Thank you for addressing the points I raised in my review. Given your response I consider Q1, Q3 and Q4 as resolved, the respective points are much clearer in the revision.
> >
> > I appreciate that you included a synthetic-supervised variant of Motion Grouping in the evaluation. However I still have concerns with the evaluation. I agree that "that the multi-object synthetic generation pipeline is also one of the important contributions made by [y]our work." In this sense, one could phrase my concern as a missing ablation study that separates the influence of the synthetic training strategy from the model architecture.
> >
> > One important experiment in this direction would be to train existing, supervised architectures on your synthetic data and compare their transfer performance to your model. A couple of methods would be interesting, for example, Mask R-CNN with RGB/optical flow input ([1]) or any of the supervised methods you compare to. For the single object case, also motion based binary segmentation models could be included (e.g., [2]). In principle, any model that is currently trained with supervision on the target dataset could be trained on the synthetic data instead. If their transfer performance is comparable, the main performance improvement is due to the synthetic training data. If not, the proposed model architecture would be shown to be more suitable for sim2real transfer.
> >
> > [1] https://openaccess.thecvf.com/content_ICCVW_2019/html/HVU/Dave_Towards_Segmenting_Anything_That_Moves_ICCVW_2019_paper.html
> > [2] https://openaccess.thecvf.com/content_ICCV_2017/papers/Tokmakov_Learning_Video_Object_ICCV_2017_paper.pdf

---

> > > ### Author Response · Authors · 2022-08-08
> > > **Reply to “Re: Response to Reviewer FcJQ”**
> > >
> > > Thank you for the suggestions. We have conducted an additional ablation study by adopting the Mask R-CNN architecture as suggested. To make a fair comparison, we here train with **only flow** input, because (i) our OCLR model is also trained on synthetic flows; (ii) our proposed Sim2Real methodology mainly focuses on the flow modality owing to its smaller synthetic-real domain gap. Our synthetic pipeline is therefore primarily targeted to simulate motions of objects and yield synthetic flows.
> > >
> > >
> > > The table below summarises the multi-object segmentation performance of three models that are trained directly on a limited amount of real data (DAVIS2017-motion for the first three rows) or on synthetic flows only (Syn-train for the last three rows). Motion Grouping (Sup.) stands for a supervised version of Motion Grouping. All hyperparameters and architectures in Mask R-CNN follow the default settings in the original paper, with a ResNet-50-FPN backbone trained from scratch.
> > >
> > >  |Model |Inputs| Supervision| Training dataset | No. of training frames |DAVIS17-motion IoU|
> > > | :--------: | :--------: | :--------: |:--------: |:--------: | :--------: |
> > > |Motion Grouping (Sup.) | Flow | Real | DAVIS17-motion | 4.2k | 32.7 |
> > > |Mask R-CNN |Flow | Real| DAVIS17-motion | 4.2k | 40.3 |
> > > |OCLR| Flow | Real | DAVIS17-motion | 4.2k | 42.8 |
> > > |Motion Grouping (Sup.) | Flow | Synthetic |Syn-train | 138.2k | 44.9 |
> > > |Mask R-CNN | Flow | Synthetic |Syn-train | 138.2k | 50.4 |
> > > |OCLR| Flow | Synthetic|  Syn-train | 138.2k | **54.5** |
> > >
> > >
> > > It can be observed that: (i) Our OCLR outperforms both benchmark models under different supervisions, particularly in synthetic-supervised results. When visualising the qualitative results, we found that Mask R-CNN demonstrates inferior performance in comparison to OCLR, particularly when there are noisy optical flows, temporally stationary objects and heavy object deformations, while in contrast, OCLR is designed with the ability to infer amodal mask, thus to handle situations with occlusion happening; (ii) Motion Grouping originally designed for self-supervisions does not perform well when direct supervision is applied; (iii) Compared to supervisions provided by a limited amount of real data, scalable synthetic supervisions lead to general performance improvements.
> > >
> > > Regarding other RGB-flow models that you have suggested, though introducing RGB information would be unfair for comparisons, we agree that training some of these models on our synthetic dataset could be useful as extra baselines. However, we are not able to train these models during the time allowed for this response.
> > >
> > > We have also added this ablation study to Section E.5 in the revised version of the supplementary material.

---

### Official Review · Reviewer_zSVP · 2022-07-11

**Rating:** 5
**Confidence:** 4
**Soundness:** 3 good
**Presentation:** 4 excellent
**Contribution:** 4 excellent

**Summary:**

**Problem**: The paper addresses the problem of discovering and segmenting multiple foreground objects in videos without using supervision.

**Solution**: The proposed solution involves training on synthetic data and only leveraging optical flow (not RGB) to facilitate easier sim2real generalization. The paper introduces a novel model architecture which tasks as input a sequence of optical flow frames and produces $K$ amodal segmentation masks each associated with an estimated depth ordering. These amodal segmentation maps can be combined using the ordering information to form the final estimated multi-object segmentation map. The architecture involves a transformer based model inspired from DETR which uses K learned queries to produce the K unique outputs.



**Questions:**

- Please discuss the question on unsupervised vs synthetic supervised presented in the Weakness above which is my only major concern.

**Limitations:**

Yes, the limitations have been discussion fairly well.
There is no discussion on the societal impact.

**Strengths And Weaknesses:**

## Strengths

### Writing
The writing of the paper is very clear and easy to follow. The paper is organized very well to facilitate easier understanding of the details of the model and proposed data generation pipeline. The details of the proposed model is presented in a concise manner covering most necessary aspects. The paper includes detailed discussions of the pros and cons of most design choices. These discussions make it much easier to follow and verify the claims.

### Proposed Model
The paper proposes a novel architecture that is interesting and might be applicable to a wider range of domains. The model makes several interesting design choices:
- The choice of leveraging optical flow only from synthetic data is not novel and has been used in other domains. However, the application to the problem of multi-object segmentation is unique.
- The model goes beyond segmenting objects in individual frames and produces *amodal* segmentation maps for each frame. This is an interesting design choice that seems to be very effective (see more about this in weakness below).
- The core of the model is heavily inspired from DETR, but the ideas of using this architecture to estimate layer depth and amodal  masks is still novel and interesting. In addition to modifying and adopting the objective of DETR for estimating the masks, the paper introduces a layer ordering loss to accurately estimate the depth order of each object.

### Synthetic Data Generation
The data generation pipeline proposed in this work is novel. The utility of this data beyond the task of multi-object segmentation is unclear. However, I believe this pipeline could be adopted by other researchers in this domain.

### Evaluation
The experimental evaluation in this work exhaustively covers standard benchmarks and the necessary ablative studies to verify the claims.
Across two tasks (single object and multi-object segmentation), the proposed model outperforms existing unsupervised learning methods. The ablative studies show the benefits of using amodal segmentation maps as the intermediate output which is a key design choice of the proposed model.

## Weakness

### Supervision and Comparisons
- The paper claims to be an **unsupervised** method for video object segmentation. I'm not sure if this is true based on the conventional usage of the term unsupervised. The proposed method uses synthetic supervision. This can obviously be corrected in the text. However, the bigger issue is the comparisons to existing work. If this work were to be categorized as a supervised learning method, the single object results are only as good as other supervised learning methods and in the multi-object case there is no comparison to supervised learning methods.
- Since the proposed method uses synthetic supervision, it is also *somewhat* unfair to compare to a model trained with real-world human supervision. But I think it is at least important to demonstrate the benefit of synthetic data *i.e.* scalability. Since generating synthetic data is not expensive, if the proposed model can scale in performance with volume of synthetic data and outperform supervised methods that use limited supervision, that would be a compelling result for adopting the proposed model.

### Amodal Evaluation
One of the interesting aspects of the proposed model is the choice of producing amodal maps as the intermediate step. However, the evaluation of this output is limited to the ablation of comparing to a model using modal maps. It would have been interesting to see how well this model performs on the amodal segmentation task.
See the following for evaluation protocol:
>Zhan, Xiaohang, et al. "Self-supervised scene de-occlusion." Proceedings of the IEEE/CVF Conference on Computer Vision and Pattern Recognition. 2020.

> Xiao, Yuting, et al. "Amodal segmentation based on visible region segmentation and shape prior." Proceedings of the AAAI Conference on Artificial Intelligence. Vol. 35. No. 4. 2021.

---

> ### Author Response · Authors · 2022-08-02
> **Response to Reviewer zSVP**
>
> Thanks for your review. Please see our comments below specifically addressing the weaknesses and questions.
>
>
> **Q1: Question on unsupervised vs synthetic supervised methods**
>
> A1: Please refer to the response to all reviewers, where we explain our reasons for adopting the term “unsupervised” and comparing our results to other unsupervised methods.
>
>
> **Q2: ''I think it is at least important to demonstrate the benefit of synthetic data i.e. scalability. Since generating synthetic data is not expensive, if the proposed model can scale in performance with volume of synthetic data and outperform supervised methods that use limited supervision, that would be a compelling result for adopting the proposed model.''**
>
> A2: Thank you for the suggestion. We agree that it would be interesting to investigate the relationship between the amount of synthetic data and final performance. We have scaled the training set as much as possible in the rebuttal period (see the table below), and will go further for the camera ready. The table shows the performance of models trained on synthetic data with frames increased from 43.2k to 138.2k.
>
> | Training dataset   | No. of training frames | DAVIS16 IoU | DAVIS17-motion IoU|
> | :--------: | :--------: | :--------: |:--------: |
> | DAVIS17-motion | 4.2k | 66.7 | 42.8 |
> | DAVIS17-motion | 4.2k (with aug.) | 69.4 | 45.3 |
> | Syn-train subset | 43.2k | 69.6 | 51.2 |
> | Syn-train subset | 69.1k | 70.2 | 51.5 |
> | Syn-train subset | 115.2k | 71.2 | 53.7 |
> | Syn-train | 138.2k | **72.1** | **54.5** |
>
> The Table above presents two messages: (i) with an increased amount of synthetic data, our model performance gets consistently improved; (ii) while training our OCLR model directly on the DAVIS17-motion training split, referring to  **supervised learning** with **manual annotations**  (“with aug.” indicates augmentations on input optical flow including random cropping, rotations, dropouts, jittering, etc.), the performance is inferior to that trained on synthetic data. Both results demonstrate the benefit of our synthetic data, especially its scalability.
>
>
> **Q3: ''It would have been interesting to see how well this model performs on the amodal segmentation task.''**
>
> A3: It’s a very interesting suggestion. Currently, we cannot carry out a comparison since the OCLR model takes optical flow frames as the only input, whereas existing image-based amodal methods and benchmarks use appearance information for amodal predictions. However, it will be very valuable in future when we finetune on real data, as amodal predictions could then provide the amodal masks for training the OCLR model. For the moment, we will add the references to the related work, and also conduct amodal comparisons when introducing the RGB modality in future work.
>
>
> **Q4: ''There is no discussion on the societal impact.''**
>
> A4: Please refer to Section H in the supplementary material for discussions on potential social impacts and ethical guidelines.

---

### Author Response · Authors · 2022-08-02
**Response to all reviewers**

We thank all reviewers for their insightful and helpful comments. We appreciate the comments on the novelty of our proposed approach, and we have tried to address the concerns and weaknesses in the individual responses. Relevant corrections and requested additional results can also be found in the revised version of the main text and supplementary material, where all modifications are labelled in **orange** for visibility.

We summarize the main changes made in our revised paper as below:
- Replace the term “unsupervised” with more accurate descriptions such as *“synthetic-supervised without any human annotations”* and *“unsupervised on real data, though supervised on synthetic flows”*.
- Indicate clearly in Table 2 and Table 3 that our method is synthetic-supervised without any human annotations.
- Add an ablation study on the scalability of model performance (Section E.4, Table E7 in the supplementary material).
- Add an ablation study on choices of optical flows (Table E5 in the supplementary material).
- Add the experiments for training the Motion Grouping model with our synthetic data, and report results on DAVIS2017-motion in Table 3.
- Clarify some derivations in modal mask generation (e.g., definitions of $\alpha^0$) in Section 2.1.
- Owing to limited space, some references worth to be mentioned as suggested will be added to the related work in the camera-ready version (with one additional page allowed).

**Q1: Definition of “unsupervised” and comparisons with other unsupervised methods**

The use of the term “unsupervised” was raised by Reviewer zSVP and FcJQ, and we give a common response here.

A1: We agree with the reviewers’ comments that it is not accurate to refer to our method as only “unsupervised”. To avoid misunderstanding, we decide to use more precise descriptions such as *“synthetic-supervised without any human annotations”* and *“unsupervised on real data, though supervised on synthetic flows”* in the revised draft.

We used the term “unsupervised”, and had comparisons to “unsupervised” methods, as this term is commonly used in the video benchmarks that we evaluate on (i.e., DAVIS16, DAVIS17), and we follow the definitions there, for example,

- In the original DAVIS16 paper [1], “unsupervised” is defined as *"They do not require any manual annotation and do not assume any prior information on the object to be segmented.”*

- For the DAVIS17 dataset, the term “unsupervised” is simply defined to differentiate from “semi-supervised” video segmentation, where the object segmentation is provided in the 1st frame, essentially, this is tracking. This definition persists even in the 2019 DAVIS challenge benchmark paper [2]: *“In the literature, several datasets for unsupervised video object segmentation (in the sense that no human input is provided at test time) have been proposed [37], [38], [39], [40], [41], [42] often interpreting the task in different ways”*

Our method does not involve any human annotations during both training and test time, which is consistent with the definitions above.

However, we have further clarified the definition in the revised version that we use synthetic supervision and apply our method on real datasets without any human annotations.






### Reference
[1] Perazz et al. “A Benchmark Dataset and Evaluation Methodology for Video Object Segmentation”. In *CVPR*, 2016.

[2] Caelles et al. “The 2019 DAVIS Challenge on VOS: Unsupervised Multi-Object Segmentation”. *arXiv:1905.00737*, 2019


### Reference in the quotation
[37] M. Narayana, A. Hanson, and E. Learned-Miller, “Coherent motion segmentation in moving camera videos using optical flow orientations,” in The IEEE International Conference on Computer Vision, 2013.

[38] P. Ochs, J. Malik, and T. Brox, “Segmentation of moving objects by long term video analysis,” in IEEE transactions on pattern analysis and machine intelligence, 2014.

[39] P. Bideau and E. Learned-Miller, “Its moving! a probabilistic model for causal motion segmentation in moving camera videos,” in CoRR,abs/1604.00136, 2016.

[40] F. Galasso, N. Shankar Nagaraja, T. Jime ́nez Ca ́rdenas, T. Brox, and B. Schiele, “A unified video segmentation benchmark: Annotation, metrics and analysis,” in The IEEE International Conference on Computer Vision, 2013.

[41] F. Li, T. Kim, A. Humayun, D. Tsai, and J. M. Rehg, “Video segmentation by tracking many figure-ground segments,” in The IEEE International Conference on Computer Vision, 2013.

[42] P. Bideau and E. Learned-Miller, “A detailed rubric for motion segmentation,” in arXiv:1610.10033, 2016.

---

### Meta-Review · Area_Chair_NfnN · 2022-08-27

**Recommendation:** Accept
**Confidence:** Certain

**Metareview:**

This paper uses synthetic data to train a CNN + transformer architecture for amodal object segmentation from optical flow input.  The model architecture can be viewed as an adaptation of DETR [12] to a different task.  Reviewer ratings lean positive, although there are concerns about experimental validation, as the combination of training regime (using synthetic data) and input modality (optical flow) does not match that of other methods tested on the same datasets; the proposed OCLR system outperforms self-supervised methods, but falls behind the state-of-the-art systems trained on real data, while using different training resources than either class.  The author response partially alleviates this ambiguity, with an additional ablation study comparing to an optical flow based Mask R-CNN model trained on synthetic data.

**Award:**

No

---

### Decision · Program_Chairs · 2022-09-14

Accept